



# Measuring frequently during peak soil N2O emissions is more important than choosing the time of day to sample

Jordi T. Francis Clar[1], Robert P. Anex[1]

[1]Department of Biological Systems Engineering, University of Wisconsin, Madison, 53706, USA

*Correspondence to*: Jordi T. Francis-Clar (francisclar@wisc.edu)

**Abstract.** Manual gas sampling from static soil chambers is commonly used to measure the flux of nitrous oxide ($N_2O$) from soil. Because manual sampling is labour intensive, sampling frequencies are often insufficient to fully capture daily variability of $N_2O$ soil flux, which compromises the accuracy of estimates of daily and cumulative emissions. Knowledge of the diurnal fluctuation of $N_2O$ flux has been used to choose a flux sampling time that maximizes the accuracy of $N_2O$ flux estimates and

thereby reduces the required frequency of flux measurements, but results of previous studies have been inconsistent. We analysed $N_2O$ soils emissions measured quasi-continuously over three years from a highly fertilized (> 200 kg N ha$^{-1}$) maize system grown in southern Wisconsin, USA. This is the first study of $N_2O$ flux temporal variability that includes multiple difficult-to-measure peak emission events ("hot moments") and estimates the relative contribution of hot moments to cumulative emissions. Analysis of diurnal fluctuation in $N_2O$ flux was performed using all measured data ($\approx$ 22,000 fluxes) as

well as using subsets of the data grouped by flux magnitude. The relationship between the observed hourly average flux and the mean daily flux was assessed using linear regression. Results show that diurnal variation in $N_2O$ soil flux was closely associated with normalized flux size. During low emission periods, $N_2O$ soil fluxes exhibited a diurnal pattern such that $N_2O$ flux measured at particular times of day, "Preferred Measuring Times" (PMTs), were not significantly different from the mean daily flux. During high emissions periods $N_2O$ flux did not exhibit a diurnal pattern and there was no PMT. High emissions

periods included difficult-to-measure hot moments that did not exhibit a PMT and contributed up to 50% of the cumulative emissions. We conclude that in order to accurately measure soil $N_2O$ flux in this type of system, it is necessary to sample frequently, particularly during peak flux events, and that constraining sampling to particular times of day provides little benefit.

## 1 Introduction

Understanding the patterns of nitrous oxide ($N_2O$) emissions from agricultural soils is a priority in the context of mitigating

global warming. $N_2O$ is a long-lived greenhouse gas (GHG) with a global warming potential 289 times that of carbon dioxide (IPCC, 2011) and its breakdown in the atmosphere is a major source of stratospheric NO which destructively reacts with the stratospheric ozone layer. Agriculture is estimated to contribute around 80% of global anthropogenic $N_2O$ emissions, more than half of which comes from agricultural soils (Syakila and Kroeze 2011) and the atmospheric concentration of $N_2O$ is increasing at 0.6 – 0.9 ppbv yr$^{-1}$ (WMO, 2014).



Measurement of N$_2$O soil fluxes is most commonly based on the use of small flux chambers (Pattey et al. 2007). The principle of this measurement technique is to place a sealed chamber over the soil surface and measure the change in N$_2$O concentration in the chamber over a period on the order of one hour by taking several samples of the chamber headspace gas. Due to soil heterogeneity, obtaining flux estimates with acceptable accuracy requires a large number of chamber replicates (Groffman et al., 2009). The combination of sampling over a long chamber closure time and the need for a large number of chamber replicates makes manual sampling very time and labor intensive. Limits on resources available for sampling campaigns often require that a single N$_2$O soil flux estimate represents the flux over an extended period, ranging from 24 hours up to as long as several weeks. This makes choosing a flux measurement that accurately represents the average soil flux during the interval between samples very important.

Nitrous oxide fluxes in soils are the result of complex biological processes which, while linked to a wide range of physical and chemical factors, are strongly influenced by soil temperature (Maag and Vinther 1996). Nitrous oxide fluxes are therefore expected to follow the diurnal pattern of soil temperature, increasing during the day and decreasing during the night. When present, this diurnal pattern of rising and falling N$_2$O fluxes means that there are particular times of day at which the measured flux will not be significantly different from the mean daily flux. We will refer to these times as Preferred Measuring Times (PMTs). If PMTs exist, and can be identified, sampling at these times would increase the accuracy of soil N$_2$O flux estimates or reduce the necessary frequency of flux measurements. Unfortunately, there is considerable disagreement in the literature about the existence and timing of diurnal patterns of soil N$_2$O flux.

Cosentino, et al. (2012) measured N$_2$O fluxes in an unfertilized soybean crop in Argentina every three hours over three days using five soil chambers, resulting in a total of 120 flux measurements. They observed that N$_2$O emissions exhibited a diurnal pattern and that fluxes measured from 09:00 to 12:00 were closer to the daily mean N$_2$O emission than fluxes measured at other times. The short duration of the experiment limited the range of soil and climate conditions observed as well as the data available for analysis.

Alves, et al. (2012) studied diurnal variability of N$_2$O emissions at Seropedica, Brazil in an unfertilized native grassland and at Edinburg, Scotland in unfertilized crop land used to grow potatoes and vegetables. In Edinburg, N$_2$O rates were measured every four hours over 30 days, yielding 180 flux measurements. In Seropedica, N$_2$O rates were measured every three hours over five days from five soil chambers, yielding 200 flux measurements. An equivalent diurnal N$_2$O emission pattern was observed at both locations. PMTs were found to be from 09:00 to 10:00 and from 21:00 to 22:00. The agreement in PMTs between the contrasting locations might be the result of equivalent diurnal temperature patterns at both locations during the observation periods (Akiyama et al., 2000; Flessa et al., 2002; Williams et al., 1999).

Laville, et al. (2011) measured N$_2$O emissions every 90 minutes from 6 soil chambers in a highly fertilized maize crop in the north of France. The crop received 76 kg N ha$^{-1}$ as dairy manure slurry before planting and 54 kg N ha$^{-1}$ as UAN three weeks later. Soon after each N application and coinciding with rainfall, N$_2$O emissions spiked. Diurnal variability was studied from



a total of 864 flux measurements taken during 9 consecutive days when it did not rain between two spikes in emissions that occurred after each fertilization event. With the exception of the two emission spikes, the 9 day period analyzed had the highest and most variable daily flux of all emissions data collected. PMTs were found to be from 07:30 to 09:00 and from 18:00 to

19:30. Cumulative emissions from April 20th to September 11th computed using a single flux measurement taken each day at 08:15 were within 10% of emissions computed using the flux measurements taken every 90 minutes. It is unclear whether $N_2O$ fluxes exhibited a diurnal pattern over any part of the longer term experiment because diurnal variability of $N_2O$ fluxes was analyzed only over the 9 day period.

To our knowledge Reeves and Wang (2015) were the first to study diurnal variability of $N_2O$ emissions using a multiyear data

set. Fluxes were measured every 2.5 hours over 3 consecutive years using three soil chambers, collecting approximately 25,000 fluxes. Soil $N_2O$ emissions were measured in southern Queensland, Australia in a wheat/barley rotation with conventional tillage and stubble retention management, receiving 90 Kg of N at planting. Measured $N_2O$ fluxes were small and exhibited low variability, ranging from 2 to 140 g of $N_2O$-N $ha^{-1}$ $day^{-1}$ and rarely exceeded 60 g of $N_2O$-N $ha^{-1}$ $day^{-1}$. Diurnal patterns were observed when fluxes were higher than 20 g of $N_2O$-N $ha^{-1}$ $day^{-1}$. The authors recognized that diurnal fluctuations of $N_2O$

emissions did not follow the same pattern on different days and that diurnal flux patterns were inconsistent across experimental replicates (i.e. chambers). However, when averaged across all emissions higher than 20 g of $N_2O$-N $ha^{-1}$ $day^{-1}$, variation in $N_2O$ emissions from the daily mean demonstrated a pronounced sinusoidal diurnal pattern. PMTs were found to be from 09:00 to 12:00 and from 18:00 to 24:00. Thus 9 out of every 24 hours were classified as PMTs and these PMTs encompass all PMTs reported in other studies we have identified in the literature that observed diurnal patterns of soil $N_2O$ flux.

Machado et al., (2019) studied diurnal the variability of $N_2O$ soil emission from fields under different management practices during two periods of the year when the bulk of $N_2O$ emissions occurred. The monitored experiments were from 2000 to 2006 and during 2015 in Ontario, Canada, and the fluxes selected for diurnal variability analysis were from the 30 day period after the major spring-thaw event and the 45 day period after N fertilization or planting. During these periods $N_2O$ flux was observed to follow the pattern of the soil temperature diurnal variation at 5-cm depth and PMTs were found to be 12 out of each 24

hours, occurring from 09:00 to 12:00 and from 17:00 to 02:00. Although there was good correlation between cumulative emissions estimated from high frequency flux measurements and from the flux measurements taken during a PMT, fluxes measured during the PMTs did not always accurately estimate the mean daily flux.

Not only is there disagreement in the literature about when the PMT occurs, some studies suggest that soil $N_2O$ fluxes may not exhibit a diurnal pattern at all when fluxes are high. In agricultural systems receiving large additions of nitrogen fertilizer,

ephemeral $N_2O$ bursts often contribute to a large fraction of the cumulative emissions. The high emissions events are short lived, lasting from hours to days and occur in response to triggers such as tillage, fertilization and rainfall (Baggs, et al., 2003; Molodovskaya et al., 2012; Sehy, et al. 2003; Yanai, et al.2004). For example, Molodovskaya, et al. (2012) observed that high emissions periods representing less than 10% of the observation time contributed as much as half of the cumulative emissions from manure-fertilized alfalfa and maize fields in New York, USA. Improving our knowledge about the possible diurnal



pattern of $N_2O$ flux during high emissions periods is important because a large portion of the cumulative emissions from intensively managed agricultural systems occur during these periods.

In a series of short experiments, with measuring intervals varying from one to two hours and experimental periods ranging from one to five days, Blackmer, et al. (1982) measured $N_2O$ emission rates using one soil chamber each in maize and fallow fields in Iowa. The experiments included urea-fertilized crops at three application rates: 80, 110 and 150 Kg N ha$^{-1}$, with

measurements occurring soon after rain. Blackmer, et al. (1982) concluded that although they observed isolated and varying diurnal patterns, they could not find a single short period of time in each day that consistently yielded the smallest difference between the measured flux and the mean daily emission.

Van der Weerden, et al. (2013) measured $N_2O$ emissions between 8 to 12 times per day over 22 to 28 days from four soil chambers placed on a pasture following bovine urine fertilization. Urine was applied to 3 plots at N loadings of 486, 501 and

508 Kg N ha$^{-1}$. Three sampling seasons yielded a total of 1850 flux measurements. Only 12 days out of a total of 71 days observed across the three sampling seasons exhibited a diurnal pattern of soil $N_2O$ flux.

During a sampling campaign of 8 days, Laville, et al. (1999) measured $N_2O$ fluxes in maize plots in the south-east of France, beginning 6 days after injection of anhydrous ammonia (200 kg of N ha$^{-1}$). The maximum hourly flux measured was more than 600 g N-$N_2O$ ha$^{-1}$ day$^{-1}$ and often exceeded 175 g N-$N_2O$ ha$^{-1}$ day$^{-1}$. Laville et al. (1999) concluded that across the 8 sampling

days there was no PMT and there was very high variation in the size and timing of the observed $N_2O$ fluxes.

Measuring soil emissions during short high emissions periods is only possible with intensive, high-frequency flux monitoring, so the literature on diurnal variability of $N_2O$ fluxes during high emission periods is limited and most of the short sampling campaigns that target such 'hot periods' do not provide information about the seasonal cumulative emissions. Assessing the importance of $N_2O$ diurnal patterns during high emissions events requires capturing multiple high emissions periods during

long-term sampling campaigns.

In previous studies a diurnal pattern of $N_2O$ soil flux has not always been observed, and when diurnal patterns were observed, the resulting PMT and its duration varied, therefore it is unclear under what circumstances a single measurement taken at the PMT can be used to estimate mean daily flux. A better understanding of the diurnal variation of $N_2O$ emissions from soils requires more study using multi-year, high frequency data that include multiple high emissions periods.

The objective of this study is to evaluate the use of PMTs as a strategy to improve the accuracy of soil $N_2O$ flux estimates or reduce the necessary frequency of flux measurements in highly fertilized crop systems. In this study we used three years of $N_2O$ soil emissions measured at sampling intervals of 2 to 6 hours from highly fertilized maize grown in southern Wisconsin, USA. Diurnal patterns and PMTs were identified by computing the correlation between the measured fluxes and the mean daily flux. The identification of diurnal patterns and PMTs was performed using the entire set of calculated fluxes as well as

using subsets of flux data grouped by normalized flux size (i.e., fractional contribution to cumulative emissions). The





practicality of using PMTs to improve the accuracy of soil $N_2O$ flux estimates or reduce the necessary frequency of flux measurements was evaluated by comparing the diurnal patterns and PMTs identified in each flux size group.

## 2 Materials and Methods

### 2.1 Experimental sites

Soil $N_2O$ emissions were measured during three sampling campaigns at the University of Wisconsin Agricultural Research Station - Arlington (ARS-A) (43°17'41.2"N 89°21'28.1"W), in Columbia County (WI). To avoid interference from previous manure amendments the experiments were performed at three different sites within close proximity of one another (< 2.25 km). The soil at the three sites was a well-drained Plano silt loam soil (fine-silty, mixed, superactive, mesic Typic Argiudoll). The three sites had previously been cropped in a maize-soybean rotation with pre-planting shallow tillage (i.e., 0.1 m depth),

and with no manure application during at least the last three years.

During the first sampling campaign in 2015, flux measurements were taken from April 2nd to October 25th. Corn (*Zea mays L.*) was planted on May 13th at a rate of 86000 seeds per ha, (Renk Seed ID# RK791SSTX), the space between rows was 0.75 m. The crop received a total of 215 kg N $ha^{-1}$ in two applications: at planting and at vegetative growth stage 6 (V6). At planting, 68 kg N $ha^{-1}$ in the form of granular urea was applied in a fertilization band located 5 cm to the side and 5 cm below the seed.

Fertilization at V6 was on June 10th, at a rate of 147 kg N $ha^{-1}$ in the form of urea ammonium nitrate (UAN) 28% solution (30% urea, 40% ammonium nitrate and 30% water) applied between rows with knife injectors at a depth of 5 to 7 cm. Corn was harvested for grain on October 22nd, yielding 14.4 T of grain per ha. We will refer to this sampling campaign as 2015.

The $N_2O$ fluxes measured during the two following campaigns are part of an ongoing experiment studying the effect of manure application timing on soil $N_2O$ emissions. During these two campaigns, fluxes were measured simultaneously from two

contiguous plots, receiving manure either in mid-September (early) or mid-November (late). In each campaign, both treatments campaigns dairy slurry was applied at a rate of 65500 l $ha^{-1}$ providing between 95 to 155 kg total N $ha^{-1}$. The slurry was incorporated within 24 hours after application using a soil finisher (i.e., 0.1 m depth). Slurry application methods and rates were based on the Nutrient Application Guidelines for Crops in Wisconsin (Laboski and Peters, 2011). During both sampling campaigns corn was planted at a rate of 86000 seeds per ha, (Pionner Seed ID# P0157AMX), the space between rows was

0.75 m. At planting, 11 kg N $ha^{-1}$ in the form of granular urea was applied in a fertilization band located five cm to the side and five cm below the seed.

The second sampling campaign occurred from September 16th 2016 to July 5th 2017. We refer to this sampling campaign as 2016-2017. Dairy slurry was applied on September 15th in the early application plot and in November 16th in the late application plot. Corn was planted on May 8th and harvested on October 23rd 2017.





The third sampling campaign occurred from September 12[th] 2017 to August 22[nd] 2018. We refer to this sampling campaign as 2017-2018. Dairy slurry was applied on September 11[th] in the early application plot and in November 13[th] in the late application plot. Corn was planted on May 8[th] and harvested on October 29[th].

**2.2 N$_2$O fluxes and ancillary measurements**

Soil N$_2$O emissions were monitored using a high resolution, near-continuous flux measurement system, which comprises a
Los Gatos Research model 914-0027 N$_2$O analyzer and four automatic soil chambers (Francis-Clar et al. 2015, Anex and Francis-Clar 2015).

The ability to measure N$_2$O concentration at both high rate and precision was key to keeping deployment times short (i.e., 10 to 20 minutes) and thereby obtaining high temporal resolution flux data. In continuous flow mode, the analyzer computes in 'real time' the N$_2$O concentration of a gas stream (i.e., 100 cc min$^{-1}$) by integrating multiple laser absorption measurements
(<3 milliseconds) over a user selected averaging time. An averaging time of 20 seconds was used, yielding a measurement precision (i.e., one standard deviation) of 1/1500 of the measured gas concentration ($1\sigma < 0.2$pbb at [N$_2$O] $\approx$ 300 ppb). With these settings (sampling rate, deployment time and analytical precision) and the chamber dimensions described below, the Minimum Detectable Flux (MDF) of the system is $4.62 \times 10^{-4}$ g of N$_2$O-N ha$^{-1}$ day$^{-1}$ for deployments times of 20 minutes and $2.67 \times 10^{-4}$ g of N$_2$O-N ha$^{-1}$ day$^{-1}$ for deployment times of 10 minutes. MDFs were computed following the method of
Christiansen et al. (2015) (Francis Clar and Anex, 2018).

Four soil chambers were distributed over an area of approximately 40 m$^2$. During the 2015 sampling seasons two chambers were placed between plant rows and the other two directly on the row. During the sampling seasons of 2016-2017 and 2017-2018 two chambers were used per plot, one placed between plant rows and one directly on the plant row. The chambers were 0.20 by 0.35 by 0.25 m tall ferromagnetic stainless steel open-ended boxes pressed into the soil approximately 0.05 m. Chamber
tops were finished with a one-inch rim to accommodate a magnetic gasket mounted on the underside of the chamber lids. Lids were made of a 12.7 mm thick HDPE plate which was supported by four levers, two at each side. Each pair of levers was mounted on steel tracks attached to both sides of the soil chambers. The opening and closing movements relied on an electrical linear actuator attached to the lid and a pull-solenoid controlling the rotation of the four levers (Francis-Clar et al. 2015, Francis Clar and Anex, 2018).

After the 2016-2017 sampling season the soil chambers were rebuilt. The redesigned chamber volume was slightly larger and incorporated an improved closing mechanism. Chambers used in the sampling campaign of 2017-2018 were 0.30 by 0.30 by 0.20 m tall and the chamber lid levers were redesigned as two parallel four bar linkages.

The analyzer was connected to each of the four soil chambers with a gas path composed of two manifold valve assembles at the inlet and the outlet of the analyzer which diverted the continuous gas flow from the analyzer to the soil chambers and vice
versa via a 30-meter-long closed loop made of Chemfluor ®FEP tubing (6.35 mm OD, 0.79 mm wall). An air pump



(AP120SEEEF40C1 - Sensidyne, Inc.) circulated the chamber headspace gas at approximately 100 cm$^3$ min$^{-1}$. The transport time between the analyzer and the chamber was determined to be 5 minutes, after which the concentration of N$_2$O in the analyzer and the soil chambers differed by less than 5% (tested at N$_2$O concentrations of 0.34 ppm and 11 ppm). Chambers were vented through a coiled 15 cm long 2.36 mm ID stainless-steel tube to allow pressure equilibration within the chamber

headspace (Hutchinson and Livingston, 2001). There were no significant pressure differentials (<10 Pa) between the interior and the exterior of the chambers during operation, as measured with a barometric pressure sensor (BMP180, Bosch GmbH) and data logger (Francis-Clar et al. 2015, Francis Clar and Anex, 2018).

Synchronization between the soil chambers, the valve assembles, and the analyzer was controlled by digital logic that allowed the user to customize the sampling sequence and the duration of the flushing and sampling periods. The digital logic included

an interruption sequence triggered by an optical rain sensor that opened all chambers during precipitation events. The measurement system was flushed before each measurement to eliminate residual gas from the previous chamber sampling. During the flushing periods all chambers remained open and pumps flushed the system, including the analyzer, with ambient air for 10 minutes. Tests showed that N$_2$O concentration at the analyzer outlet differed by less than 2% from ambient after 5 minutes of flushing when the initial N$_2$O concentration in the system was approximately 11 ppm. Chamber sampling time was

set to 20 minutes, except for short periods when high flux emissions were observed. During high emission periods a 10 minute sampling time was used, yielding temporal resolutions of 12 and 18 flux measurements per chamber per day for 20 and 10 minute sampling periods, respectively.

In addition to N$_2$O flux measurements, soil temperature and moisture and weather data were recorded following Kladivko et al. (2014). Soil temperature and moisture were measured at the quarter-row position (less than 4 meters radius from the soil

chambers) every 15 minutes using five soil probes (5TM, Decagon Inc.) installed at depths of 10, 20, 40, 60 and 100 cm. Soil measurements were recorded using an em50 data logger (Decagon Inc.). Air temperature and rainfall measurements taken each 30 minutes were obtained from the Arlington-ARS Weather Station (43°17'48.0"N 89°23'03.4"W) located less than 2 km from the experimental sites.

**2.3 N2O flux estimation**

Soil N$_2$O gas flux was estimated from the change in gas concentration in the chamber headspace over time. Gas flux per unit soil area was estimated from the slope obtained by least-squares linear regression of the concentration of [N$_2$O] versus time (t) to estimate d[N$_2$O]/dt, as in Eq. (1).

$$N_2O_{flux} = H \frac{d[N_2O]}{dt} \tag{1}$$

where H is the ratio of the internal chamber volume to area of soil surface enclosed by the chamber. Flus of N$_2$O is generally

expressed in units of mole or mass of N$_2$O-N per units of area and time (e.g., mol N$_2$O-N ha$^{-1}$ day$^{-1}$ or g N$_2$O-N ha$^{-1}$ day$^{-1}$) (Parkin et al., 2003).



Use of the high-precision, cavity enhanced laser absorption spectroscopy instrument, capable of measuring near-ambient levels of $N_2O$ enabled very short chamber deployment times (<0.25 h) and use of a linear flux calculation. Estimates of soil gas flux using surface chambers tend to underestimate actual emission rates because as the concentration of $N_2O$ in the headspace

increases, the vertical concentration gradient driving diffusion of $N_2O$ into the chamber necessarily decreases (referred to as the 'chamber effect'). The error resulting from this inherent nonlinearity of $N_2O$ flux is minimized by using a nonlinear flux calculation or by maintaining a low $N_2O$ concentration in the chamber through short deployment times as done here (Venterea, et al. 2009, Parkin, et. al 2012).

Total chamber closure times used were either 20 minutes or 10 minutes, depending on flux intensity, the corresponding

effective chamber deployment (i.e., sampling) times were 15 or 5 minutes respectively, after accounting for gas transport time in the sampling system. The analyzer sampling rate was set to 20 seconds yielding an approximate precision of 0.2 ppb and recording 45 or 15 $N_2O$ concentration measurements per a 15 or 5 minute chamber deployment time, respectively. Having such a large number of measurements allowed us to reliably detect and eliminate chamber effects by testing for linearity in the flux calculation and subsampling the data when necessary to assure flux linearity. The first step in the adaptive linear flux

calculation was to estimate the flux (change in chamber headspace $N_2O$ concentration vs. time) and the corresponding coefficient of determination ($r^2$) using all data collected during the effective sampling period. If $r^2$ was smaller than 0.95, a new flux estimate (i.e., slope of $N_2O$ concentration vs. time) and corresponding $r^2$ were calculated using a subsample of the data. Subsamples were created by sequentially eliminating the last $N_2O$ concentration datum until the computed $r^2$ was larger than 0.95, with a minimum of 6 time-concentration data points. This adaptive linear flux calculation allowed us to minimize

chamber effects without compromising the precision or accuracy of the flux estimates.

**2.4 Data selection and statistical analysis**

Soil $N_2O$ flux estimates calculated from the chamber concentration measurements were screened to eliminate unreliable and *de minimis* flux measurements prior to statistical analysis. Estimated fluxes that were below the MDF corresponding to the chamber closure time (e.g. < 4.62 x $10^{-4}$ g $N_2O$-N ha$^{-1}$ day$^{-1}$ for 20 minute closure) were indistinguishable from zero flux and

were removed from the flux dataset. In addition, we screened for unreliable flux estimates resulting from occasional malfunctions of the unsupervised measurement system that occurred when a chamber failed to open or failed to close. For example, a chamber might not close or open if ice buildup blocked the chamber lid linkage during a freezing rain. A flux estimate was deemed unreliable and rejected due to failure of a chamber to close when the measured chamber $N_2O$ concentrations at the beginning and end of the sampling period were both within ± 2 times the instrument precision (0.4 ppbv)

of the ambient atmospheric $N_2O$ concentration. If a chamber failed to open, it would remain closed through a complete 2 hour cycle of sampling all four chambers, and the chamber headspace $N_2O$ concentration would be in equilibrium with the $N_2O$ concentration in the soil or very nearly so. Therefore, a flux estimate was deemed rejected due to failure of a chamber to open when the measured chamber $N_2O$ concentration at the beginning of the sampling period was greater than the ambient



atmospheric $N_2O$ concentration by +2 times the instrument precision and the chamber $N_2O$ concentration at the end of the
sampling period was within ± 2 times the instrument precision (0.4 ppbv) of the chamber $N_2O$ concentration at the beginning
of the sampling period. Ambient atmospheric $N_2O$ concentration was measured by sampling the ambient air 2 m above the
instrumentation trailer during the 5 minutes prior to chamber closure.

Capturing the daily variability of $N_2O$ fluxes is essential to testing the hypothesis that sampling at one particular time of the
day is a reasonable approximation of the mean daily flux. Consequently, only days with a minimum of 6 flux measurements
and with gaps between flux measurements of no more than 4 hours were included in the analysis. That is, only the high time-
resolution flux data were analyzed.

The '*daily flux*' of $N_2O$ at each chamber (g N-$N_2O$ ha$^{-1}$day$^{-1}$) was computed as the integral over 24 hours of the individual flux
estimates at that chamber on a specific day. '*Mean daily flux*' at a chamber was calculated as the daily flux at that chamber
divided by 24. The annual '*cumulative flux*' at each chamber was computed as the sum of the daily fluxes over a year.

The similarity between the mean daily flux and a flux recorded at a specific hour of the day was assessed through a linear
regression following the methods of Alves et al., 2012; and Cosentino et al., 2012). All estimated fluxes were binned into one
of 24, one-hour, intervals according to the hour of the day when the chamber deployment began (the '*sampling interval*'). The
common logarithm of all fluxes recorded during a particular sampling interval (the *'hourly fluxes'*) were regressed on the
common logarithm of the corresponding mean daily flux and chamber using least squares regression with zero intercept, Eq.
265   (2).

$$log_{10}(mean\ daily\ flux_{chamber,day}) = \beta_{hour} \times log_{10}(hourly\ flux_{chamber,day,hour}) + 0 \qquad (2)$$

The regression coefficient β is referred to as the '*deviation coefficient*' or simply 'β'. The statistical significance of the
difference between the mean daily flux and the flux measured during a particular sampling interval was tested (t-test, p value
< 0.05) by comparing the regression coefficient β to a value of 1.

The magnitude of soil $N_2O$ flux can be highly variable on weekly, monthly or seasonal scales due to variations in the levels of
available oxygen, nitrogen, and carbon in the soil. The mechanisms controlling the availability of these limiting resources, and
therefore the size of the soil $N_2O$ flux, are expected to vary both seasonally and with events like precipitation, tillage and
fertilization. Therefore statistical analysis was performed using all available flux estimates and subsequently using subsets of
the individual flux estimates grouped by normalized size of cumulative daily flux.

The normalized cumulative daily flux size was calculated for each chamber and each year as the percentage the annual
cumulative flux represented by the cumulative daily flux observed at the chamber. To create the data subsets using to the
normalized cumulative daily flux size we computed a new variable referred to here as '*cumulative contribution*'. *Cumulative
contribution* was computed as the result of successive additions of normalized cumulative daily fluxes that were sorted by size
in descending order. For example if the 3 top normalized daily fluxes were 2%, 1.5% and 1% the resulting cumulative



contributions for each successive normalized daily flux would be 2%, 3.5% and 4.5%. Using values of cumulative contribution as breaking points we created four subsamples of flux estimates: 75% High Cumulative Contribution (HCC), 50% HCC, 30% HCC and 50% Low Cumulative Contribution (LCC). In a HCC sample, the fluxes included in the subsample are from days that are above the cumulative contribution threshold while the fluxes included in the LCC subsample are from days below the cumulative contribution threshold. For example, the 50% HCC includes all estimated fluxes from the days included in the set

of largest daily fluxes which sum to 50% of the cumulative flux. Similarly, the 50% LCC includes all estimated fluxes from the days included in the set of the smallest daily fluxes which sum to 50% of the cumulative flux.

## 3 Results

The number of flux estimates that were below the MDF or deemed to be unreliable due to a sampling system malfunction represented 13%, 48% and 15% of the total measured fluxes during 2015, 2016-17 and 2017-18, respectively. From the

remaining 23,793 fluxes, 21,865 were estimated with high temporal resolution, accounting for 551, 373 and 1093 days of high frequency flux measurements, gathered during the 2015, 2016-17 and 2017-18 sampling seasons, respectively. In sum, high temporal resolution flux measurements represented 2017 days with an average temporal resolution of 11 fluxes per day. On average, deviation coefficients (β) were estimated from 912 measurements. The 50% and 30% High Cumulative Contribution (HCC) subsamples contained the fewest flux measurements (Fig 1, panel (d) left plots), for these subsamples β values were,

on average, computed from 55 and 27 measurements, respectively.

The β values computed for all hourly sampling periods and data subsamples (HCC, LCC, seasons and sampling campaigns) were significant (p value < 0.05); β values ranged from 0.82 to 1.1. The coefficients of determination ($r^2$) ranged from 1 to 0.78, indicating that the variability of the mean daily flux was well explained by the flux measured during each period. In general the $r^2$ values for regressions of the fluxes measured between 00:00 to 12:00 were smaller than the $r^2$ associated with

fluxes measured between 12:00 to 23:00 (data not shown).The largest variability of β values was observed at the subsampling levels of 50% and 30% HCC, for regressions of the fluxes measured between 00:00 and 09:00 (Fig 1 Panels (c) and (d), right).

When examined as a whole, the $N_2O$ flux estimates exhibited a clear diurnal pattern. Relative to the mean daily flux, $N_2O$ flux was generally lower in the morning and higher in the afternoon. The β values computed from fluxes observed during the sampling intervals beginning between 03:00 and 10:00 were greater than 1, while the values of β for the sampling intervals

beginning between 13:00 and 22:00 were less than 1 (Right panels of Fig 1, Panels (a) and (b), right and Fig 2).

The maximum β value (1.052) was associated with fluxes observed during the 07:00 sampling interval. The β values for sampling intervals after 07:00 decreased monotonically toward the minimum (0.923) in the 17:00 interval. The value of β for fluxes observed at sampling intervals beginning at 01:00, 02:00, 11:00, 22:00, and 23:00 were not significantly different from one (p value > 0.05) (Fig 1, panel (a) right).



The β values observed from the subsamples at 75%, 50%, and 30% HCC level were largest at 05:00 and smallest at 15:00; except for the subsample at 75% HCC level which exhibited a minimum β at 07:00 (Fig. 1 Panels (b), (c) and (d), right). The β values for the high cumulative contribution (50% and 30% HCC) subsamples did not follow a smooth diurnal pattern as observed in the full data set. The variability of β at the subsampling levels of 50% and 30% HCC was higher than that computed for the full data set, particularly in the sampling intervals from 00:00 to 12:00 (Fig. 1 Panels (c) and (d), right). The 95%

confidence intervals of the β values for the subsamples at 75%, 50% and 30% HCC were on average two, three and six times greater than those computed using the full set of flux estimates (Fig 1, right panels).

## 4 Discussion

To our knowledge this is the first multiyear study of diurnal variability of soil $N_2O$ emissions in highly fertilized agronomical systems in which ephemeral bursts of $N_2O$ emissions (e.g. 'hot moments') were measured. The results show that the diurnal

variability of $N_2O$ soil fluxes varies with to flux intensity.

During low emissions periods, $N_2O$ soil fluxes exhibited a diurnal pattern, in which $N_2O$ fluxes increased beginning at sunrise and decreased during the night (Fig 2). In this experiment, PMTs were during the hours beginning at 01:00, 02:00, 11:00, 22:00 and 23:00 (Fig 2). The diurnal pattern and PMTs identified during analysis of the set of all flux estimates were the same as those identified through analysis of the set of low flux estimates (Fig 1, Panel (a), Right). This is not surprising since the

low flux estimates represent approximately 85% of the total number of estimated fluxes. The tendency of $N_2O$ soil flux to exhibit a diurnal pattern in which $N_2O$ fluxes increase after sunrise and decrease after sunset and through the night has been observed by others (Table 1). Although the previous researchers have observed a diurnal pattern of soil $N_2O$ flux, in general, different studies have identified different PMTs (Table 1).

Since the diurnal pattern of $N_2O$ emissions is usually related to soil temperature (Table 1), it is expected that under different

experimental conditions (e.g. location, season, etc.), PMTs will occur at different times of the day. In fact, analysis of our flux data grouped by seasons (i.e. winter, summer, spring and fall) revealed that the diurnal pattern in which $N_2O$ fluxes increased following sunrise and decreased during the night was evident during all four seasons. The duration and timing of PMT, however, varied from season to season (data not shown).

Although the experimental conditions of the studies that have reported a diurnal pattern of soil $N_2O$ flux vary widely, the range

of $N_2O$ fluxes observed during these experiments is within the range of the set of low emissions that were observed in this study (0 to 723 g of $N_2O$-N ha$^{-1}$ day$^{-1}$), is consistent with the hypothesis that low emissions systems tend to exhibit a diurnal pattern (Table 1).

During the high emissions periods of our experiments, $N_2O$ fluxes did not exhibit a diurnal pattern and thus there was no PMT (Fig 1, Panels (c) and (d), right). In this study, high emission periods were the result of peaks of $N_2O$ flux lasting from one to

several days (Fig 1, Panels (c) and (d), left). These very high fluxes were triggered by precipitation events following





fertilization and/or soil thaw. Although high emissions periods observed in this study represent less than 6% of the total flux data, they contributed as much as 50% of total annual emissions. Peak $N_2O$ fluxes like those observed during the high emission periods in this study have been characterized as 'hot moments' or 'hot periods' which occur most frequently in highly fertilized crops and have been observed most frequently in the upper Midwest region of the U.S. (Groffman et al., 2009; Molodovskaya

et al., 2012; Wagner-riddle et al., 2007). This article is the first to analyze diurnal variability of $N_2O$ soil emissions during hot moments and hot periods from multiple years and under a range of weather conditions, and occurring following both summer fertilization and spring soil thaw. Diurnal variability studies carried out in crop systems in which peak emission events occur, support our observations during high emissions periods. Blackmer, et al. (1982) measured $N_2O$ soil emissions from highly fertilized corn systems in the Midwest of the US, at sub-daily intervals during short sampling campaigns (i.e. days); the

observed fluxes did not exhibit a PMT. Laville, et al. (1999) studied the diurnal variability of $N_2O$ soil emissions measured during a 'hot period' lasting six days and occurring after precipitation following the injection of 200 kg N ha$^{-1}$ in a maize crop in the south-west of France. During this high emission period, $N_2O$ fluxes were highly variable at the hourly scale and did not exhibit a diurnal pattern. Šimek, et al. (2010) measured $N_2O$ fluxes from a cattle overwintering area, the extremely high $N_2O$ emissions observed during soil thaw did not exhibit a diurnal pattern.

Because in the crop system studied in this article ephemeral high emissions periods do not exhibited a diurnal pattern of $N_2O$ flux and represented up to 50% of the cumulative flux, measuring $N_2O$ fluxes once a day during a PMT would not guarantee accurate estimation of cumulative flux.

In other systems in which episodic peaks of $N_2O$ emissions are not observed or are infrequent (Barton et al., 2015; Pennock, et al. 2006), measuring soil $N_2O$ flux at the PMT could possibly be an appropriate way to estimate daily flux (Reeves et al.,

2016; Reeves and Wang, 2015).

## 5 Conclusions

This is the first study in which multiyear and high temporal resolution soil $N_2O$ measurements from a highly fertilized maize system in the Midwest U.S. were used to analyze the diurnal variability of soil $N_2O$ flux. We found that in this system (i.e., cropping system, soil and climate) the diurnal variability of $N_2O$ emissions is closely related to flux intensity.

Analysis of all observed fluxes showed that soil $N_2O$ flux exhibited a diurnal pattern in which $N_2O$ fluxes were larger than the mean daily flux in the morning and less than the mean daily flux during the evening and night. As a result, $N_2O$ fluxes measured at certain times of day were not significantly different from the mean daily flux (p value > 0.05), and are thus the preferred measuring times (PMTs). In this study the PMTs were during the hours beginning at 01:00, 02:00, 11:00, 22:00, and 23:00. The diurnal pattern of soil $N_2O$ flux and the resulting PMTs observed in the full set of approximately 22,000 flux estimates

were the same as those observed in the estimates of the low emissions period fluxes (Fig 2) because in this system low emissions periods represent 85% of the total observations.

During high emissions periods $N_2O$ fluxes did not exhibit a diurnal pattern and therefore during these periods there were no PMTs. In this system, high emissions periods were observed in each year and comprised both single and multi-day $N_2O$ flux peak events representing less than 6% of the total observations, but contributed up to half of the cumulative $N_2O$ flux.

Intensively managed cropping systems like the one studied here are the source of three-quarters of total anthropogenic $N_2O$ emissions (EPA 2019). These intensively managed systems also tend to exhibit brief periods of high $N_2O$ flux that are responsible for a large fraction of total annual emissions from the soil. Emissions during these periods, however, do not follow a diurnal pattern and cannot be accurately measured with infrequent measurements that are assumed to be representative of flux during the entire sampling period. Yet it is precisely these cropping systems, the source of most anthropogenic $N_2O$

emissions and intensively managed by humans, that we have the best chance of managing to reduce emissions. As this work demonstrates for intensively managed maize in Wisconsin, USA, in order to accurately capture the patterns of soil $N_2O$ flux and accurately inventory soil $N_2O$ emissions, it is necessary to sample such systems frequently, particularly during peak flux events, and there is little benefit to trying to identify the best time of day to do that sampling.

**Data and Code Availability**

The dataset and statistical analysis supporting this results are available and open access via the MINDS@UW data repository that is supported, maintained and managed by the UW Libraries.

doi: https://doi.org/10.21231/6a7b-xy41

Permanent link: http://digital.library.wisc.edu/1793/79416

**Sample availability**

Not Applicable

**Video supplement**

No video supplement

**Appendices**

No appendices

**Team List**

Not applicable





**Author Contribution**

Jordi T Francis-Clar: conceptualization, data curation, formal analysis, investigation, methodology, project administration, software, validation, visualization and writing original draft.

Robert P. Anex: conceptualization, funding acquisition, methodology, project administration, resources, supervision, visualization and writing – review, editing.

**Competing interests**

The authors declare that they have no conflicts of interest.

**Disclaimer**

Not applicable

**Acknowledgements**

The authors are grateful to the personnel of the Arlington Agricultural Research Station. We thank our colleagues: Mark Allie, Kody Habeck, Lawrence G. Oates, Andy Larson, Todd. W. Andraski and Carrie Laboski for material and intellectual support in this research.  This material is based upon work that was supported by the National Institute of Food and Agriculture, U.S.

Department of Agriculture, Hatch projects under accession numbers 1001805 and 1009785.

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

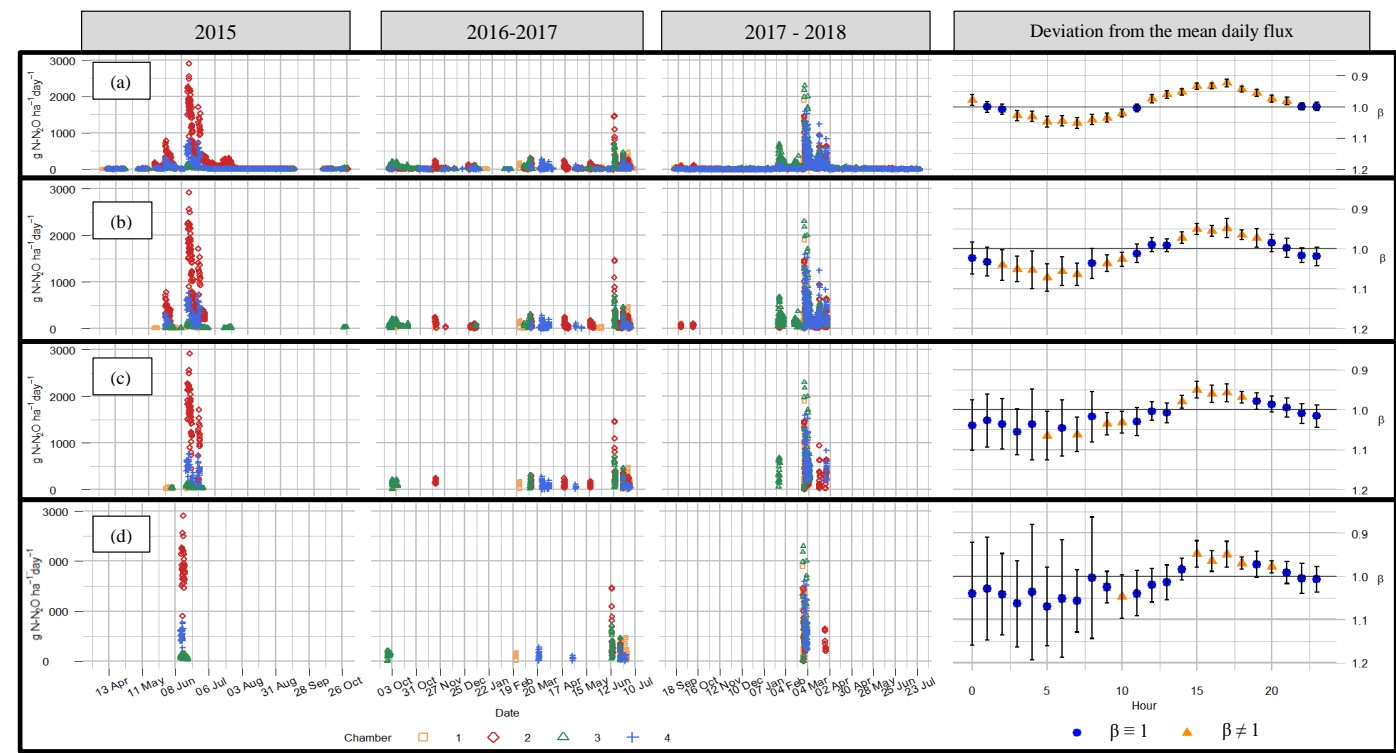

**Figure 1.** N₂O fluxes from three sampling campaigns (left columns) and the diurnal deviation of the mean hourly flux from the mean daily flux (far right column). The rows (a, b, c and d) are the fluxes and diurnal deviation of the largest fluxes that account for a cumulative contribution of 100%, 75%, 50% and 30% of total emissions, respectively. These account for and 100%, 15%, 6% and 3% of the total observations, respectively. The deviation coefficient β was computed by least squared regression of the logarithm of the mean daily flux on the logarithm of the flux measured during certain hour of the day $log_{10}(mean\ daily\ flux_{chamber,day}) = \beta_{hour} \times log_{10}(hourly\ flux_{chamber,day,hour}) + 0$. Blue circles indicate no significant difference between β and 1 (p value >0.05). Vertical bars are 95% confidence intervals. Variability and uncertainty of the deviation coefficient β is highest for the largest fluxes (shown in row (d): 30% of the total flux from 3% of the total observations).



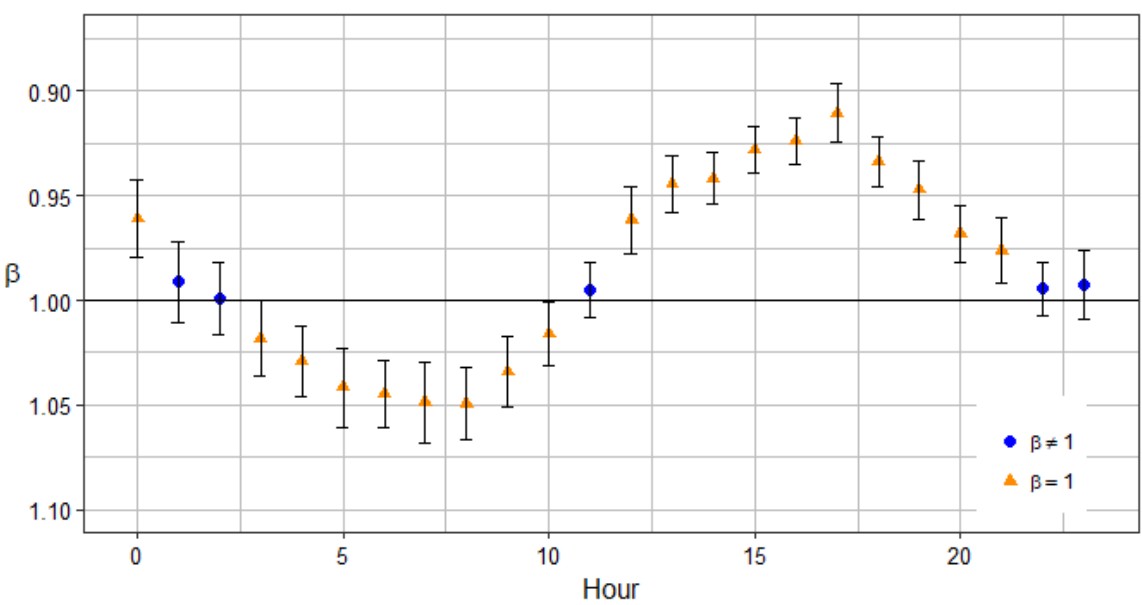

**Figure 2: Diurnal deviation of average hourly flux relative to the mean daily flux for the smallest fluxes. The diurnal deviation coefficient, β, was computed by least squares regressions of the logarithm of the mean daily flux on the logarithm of the flux measured at a certain hour of the day** $log_{10}(mean\ daily\ flux_{chamber,day}) = \beta_{hour} \times log_{10}(hourly\ flux_{chamber,day,hour}) + 0$ **. Blue circles indicate no significant difference between β and 1 (p value < 0.05), and vertical bars are 95% confidence intervals. The β values shown are computed for low flux periods (the smallest fluxes which contributed to 50% of the total emissions, representing 85% of the total observations) are relatively consistent across hours of the day and the associated uncertainties are also relatively uniform and small.**





**Table 1. Summary of literature reporting a diurnal pattern of N₂O soil emissions and PMTs**

| Reference | Flux range (g of $N_2O$-N ha$^{-1}$ day$^{-1}$) | Reso-lution | Number of days observed | Diurnal pattern observed | PMTs (Preferred Measuring Times) | Corre-lation with temp. | Soil cover | Location |
|---|---|---|---|---|---|---|---|---|
| **This study** | **Low emissions (LCC 50%)** 0 to 723.16 (median = 4.3) | 2.5 h | 1909 (LCC 50%) | **Yes** | 01:00 to 02:00, 11:00, 22:00 to 23:00 | Yes | Maize | Arlington, Wisconsin |
| Akiyama, et al. 2000 | 0.6 - 1 | 4 h | 6 | Yes | 08:00 to 12:00 | Yes | Carrots | Tsukuba, Japan |
| Flessa et al. 2002 | 1.8 - 6 | 12 h | 8 | Yes | 08:00 to 12:00 | yes | Potato | Munich, Germany |
| Williams, et al. 1999 | 0.36 – 0.4 | 2.67 h | 3 | Yes | 12:00 to 14:40 | Yes | Perennial grass | Cumbria, UK |
| Jantalia et al. 2008 | 2.5 – 33.5 | 3 h | 3 | Yes | 07:00 to 10:00 | Yes | Perennial grass | Paso Fundo, Brazil |
| Denmead 1979 | 2.3 – 3 | < 1h | 2 | Yes | 09:00 to 12:00 | Yes | Perennial grass | Camberra, Australia |
| Rosa et al. 2012 | 1.5 – 3.5 | 3 h | 3 | Yes | 09:00 to 12:00 | Yes | Soybeans | Buenos Aires, Argentina |
| Laville, et al. 2011 | 17.3 – 103.4 | < 1 h | 9 | Yes | 07:30 to 09:00 and 16:00 to 19:30 | Yes | Maize | Gignon, France |
| Alves et al. 2012 | 0 – 10.8 | 3 and 6 h | 35 | Yes | 09:00 to 10:00 and 21:00 to 22:00 | Yes | Perennial grass/ Potato | Seropedica, Brazil / Edinburgh, Scotland |
| Reeves and Wang 2015 | 20 – 140 | < 1h | 3 yrs. | Yes | 09:00 to 12:00 and 21:00 to 24:00 | Yes | Wheat/ barley | Queensland, Australia |
| Reeves, et al. 2016 | 0 - 550 | 2-3 h | 1yr | Yes | 09:00 to 12:00 and 21:00 to 24:00 | Yes | Sugarcane | Queensland, Australia |
| Savage, et al. 2014 | 0 – 4.8 | 1 h | 74 | Yes | 09:00 to 10:00 | Yes | Forest wetland | Bangor, Maine |
| Maljanen, et al. 2002 | 0 - 24 | 6 h | 38 | Yes/No | NA | Yes | Multiple | Eastern Finland |
| Shurpali et al. 2016 | 0 - 350 | 1 h | 214 | Changing with flux intensity | NA | Yes / No | Perennial grass | Eastern Finland |





| Machado et al., 2019 | -16 - 496 | 0.5, 1, 2 and 4h | 2280 | Yes | 09:00 to 12:00 and 18:00 to 02:00 | Yes | Corn, soybean, wheat | Ontario, Canada |
|---|---|---|---|---|---|---|---|---|