# Peer review of "Measuring frequently during peak soil N2O emissions is more important than choosing the time of day to sample"

_Biogeosciences, 2019_

## Referee Comment (RC1) · Anonymous Referee #1 · 19 Nov 2019

General comments: Authors state several times that this is the first study of N2O flux temporal variability that includes several hot moments. Which isn't really true, is it? There is Luo et al. 2012 (Decadal variability of soil CO2, NO, N2O, and CH4 fluxes at the Hogwald Forest, Germany), and a lot of other studies from that same study site that show temporal variations in N2O. There are also numerous papers from Australia (e.g. Barton et al. 2007 Nitrous oxide emissions from a cropped soil in a semi‐arid climate) – although, to be fair, the Barton paper did not really experience what could be called "hot moments". And there is also the Machado et al. 2019 paper (as cited in the current manuscript) that also measured temporal variability during periods that included "hot moments".

[Figure]

Also, I don't agree that constraining sampling to particular times of the day provide little benefit. Generally, researchers will be sampling during regular working hours (i.e. between 9 am and 5 pm), during those hours, sampling before 10 am will underestimate fluxes (for 50% of annual flux – i.e. see figure 1c), while sampling after 2 pm will overestimate fluxes. So your preferred sampling time should be between 10:30 to 13:30. No? Even figure 1d makes a strong case for not sampling in the afternoon (between 15:00 and 18:00), because it will overestimate fluxes. That being said, I do agree with you that frequency of sampling during hot moments is more important than what time the sampling took place during those hot moments.

So, realistically, I don't think that your conclusions are actually substantiated by your own data. I think that your data still backs up previous research that suggests avoiding the afternoons when sampling for determination of N2O fluxes. Not the most novel conclusion, but I think it is still worthwhile.

Also, I would like to see a bit more in the discussion about why diurnal patterns are less relevant during periods of high emissions. There is very little on what mechanisms of processes actually drive this. In Figure 1c and 1d, there is much higher variability between 0 and 8h (compared with the rest of the day). Any ideas why this might be? Is there less production? Or is it related to climatic conditions at night?

Finally, try to avoid paragraphs that consist of 1 sentence (e.g. lines 159-161).

Specific comments:

Lines 36-38. The topic of sampling frequency and quantifying what it means in terms of uncertainty for your cumulative estimate has been covered well in the paper by Barton et al. 2015 (see your own citation list), and should probably be cited here.

Line 46: I think it would be worthwhile to cite your Table 1 here, because the table does a good job of summarizing some of the literature on timing and existence of diurnal patterns in N2O fluxes. Although, it doesn't seem like there is that much disagreement.

Pretty much all of the studies summarized in Table 1 (11 of the 12 that suggest a PMT) suggest avoiding sampling during the afternoon. That is pretty good consensus in my opinion.

M&M section is very thorough. Nice.

Line 214: "Flux" not "Flus".

Line 239: wouldn't it make more sense to report your MDF as a flux per hour? Rather than per day? You are measuring flux rates based on 20 min deployment times and are calculating your daily fluxes by integrating the individual flux measurements for that particular day. So it is possible to have some fluxes below the MDF and others above the MDF on the same day.

Line 240: I'm not sure why these were removed from the dataset. I would have retained them. Also, it is not clear what you did with these values. Whether you assigned them a value of 0 or just deleted them could create differences when calculating cumulative fluxes via integration.

Line 266: the "+0" is unnecessary.

Line 275: "percentage of the annual".

Results:

Line 289: what happened during 2016-17? Having to throw out 48% of the fluxes is not good (or were these removed because they were below MDF?). I think it might be better to differentiate when fluxes were thrown out because of bad data and when they were thrown out because they were below MDF.

Discussion:

Line 318: Are you sure that this is the first? Doesn't the Machado et al. 2019 paper measure diurnal variability in a fertilized agronomic system with hot moments?

Line 319-320: I am not sure what this sentence is trying to say. Diurnal patterns of N2O emissions vary due to flux intensity? Or something like that? It would be good to add a short discussion why that may be.

Line 334-337: a single sentence is not a paragraph. Also, I have read this a few times, and it is really hard to follow. Please re-phrase.

Line 345-347. I'm pretty sure that the Machado paper that you cite analyzes diurnal variability of N2O during hot moments from multiple years, under a range of weather conditions and following summer fertilization and spring thaw periods. So I don't think that this statement is correct.

Line 355: "did not exhibit", rather than "do not exhibited".

Line 356: this is partly true. While measuring during a PMT may not guarantee accurate estimation, measuring during the afternoon would almost certainly cause biased estimates (see figure 1d).

Conclusion:

Line 362: You keep saying that you are the first, and yet you have cited another manuscript that has also done this.

Line 372: In figure 1c (50% of flux), there is a pattern during daylight hours (underestimate during early morning and late evening, and overestimate during the afternoon), but not at night. So unless you plan on sampling at night, there are still PMTs. Also, any idea why there was so much more variability in flux measurements made at night?

Line 384: According to your data, there still seem to be periods that should be avoided (i.e. afternoons). This should be mentioned.

Figure 1: Looking at the number of points, I would guess that each point equals one hourly flux. So please change the Y axis to "g N2O-N ha-1 hr-1" (or per m2) rather than per day.

References:

Please go through and check the formatting of the references. There are many where the formatting is incomplete.

---

## Referee Comment (RC2) · Anonymous Referee #2 · 20 Nov 2019

The authors present observations of N2O emissions from an agricultural research station in Wisconsin over three different seasons at high temporal resolution using a Los Gatos N2O analyzer and automatic chambers in an effort to understand diurnal variability of N2O emissions and whether certain times of day are representative of daily emissions. Emissions were measured from three different plots: in one from April through Oct 2015, one from September 2016 to July 2017, and one from September 2017 through August 2018. They integrate daily fluxes based on a minimum of 11 observations, and then rank the daily fluxes from largest to smallest. They bin these ranked daily observations based on size into 4 or 5 largely overlapping bins: observations amounting to 25% (or possibly 30%) of annual fluxes, observations amounting

to 50% of annual fluxes (or possibly 25%), and all observations. An additional bin contains the observations ranked from smallest to largest that amount to 50% of annual fluxes. In the datasets containing larger proportions of the observations, there is a clear and relatively well constrained diurnal variability. The large 50% and 25% (or 30%) bins - which represent the largest fluxes ('hot moments') - exhibit high variability and lack clear diurnal variability. The authors conclude that relying on a representative sampling time will not accurately estimate emissions during these high flux dates.

The authors have done a commendable job of collecting high temporal resolution measurements across three separate years – it is a very nice dataset, and I very much appreciate the effort undertaken to obtain it. That said, I have several major concerns.

1. I leave the question of editorial suitability to the editor, but I could use a little more convincing that this manuscript wouldn't be better suited to a journal more focused on agriculture, given the manuscript's narrow focus on questions of sampling representativeness of fertilized fields. With respect to novelty, previous studies have similarly found that N2O fluxes often do not exhibit diurnal variability, particularly when fluxes are low or high (e.g., Laville et al 1999, Van der Weerden et al. 2013), although to say that these particular studies are much shorter in duration would be an understatement, and the long-term dataset included here is certainly of value.

Incidentally, one interpretation of the results would seem to be that PMTs could be the optimal design in instances where sub-daily sampling is not possible: if PMTs accurately reflect daily emissions during the majority of sampling days, it makes sense to use them. If on high emission days there is no diurnal variation, measuring during a PMT is as good a time as any. Perhaps the real message might be that effort should be made to sample hot moments in higher than daily temporal resolution. A question I'm left with is: how does variation in the sampling timing and frequency during hot moments affect annual flux estimates (e.g., comparing a single daily measurement at the PMT to varying sets of additional, sub-daily measurements)? In other words, some quantification of the downside of using a PMT would be very interesting, and germane

to the central question of this study.

2. The analyses presented here are focused on the question of sampling representativeness based on an empirical evaluation of when diurnal variation does and does not occur. It does not attempt to explore why and under what conditions diurnal variation breaks down: there's almost no interrogation of process or mechanism. Understanding the degree to which PMTs are representative of daily fluxes is a worthwhile goal, but the presentation here feels thin without substantive investigation of why PMTs are or are not representative. There are many data here that were collected but don't appear to figure at all in the analysis and are not presented as results, in particular depth-resolved soil temperature and moisture observations (Table 1 informs us that some kind of temperature/N2O relationship existed, but no details on that relationship are presented). Seasonal variation in the duration and timing in PMTs is referred to in passing but not actually presented. There is no consideration of soil C or N. There are a lot of questions I was curious about while reading the manuscript. For example, does diurnal variation in high fluxes depend at all on whether the pulses are related to fertilizer applications, freeze-thaw events, precipitation events, or some other cause? Do organic and inorganic fertilizer applications affect diurnal variation differently (through effects on soil O2 and organic C availability to denitrifiers)? How does (seasonal) variation in the range of diurnal temperature variation affect diurnal variation during high flux events? Even with the three-year data set there may not be enough replication of events to answer some of these questions statistically, and the organic v inorganic question can't be analyzed statistically with the current experimental design, but I would think some quantitatively-informed discussion would be possible, and could be a good way to take advantage of this very nice dataset.

3. I have a number of concerns with the binning approach used.

First, the description of the normalized cumulative daily fluxes in line 274 and following is not very clear and could be done much more simply: I'm not sure it is necessary to define a new concept here (It's also not necessary to normalize, especially since

[Figure]

the normalization is not actually carried through in the analysis: the results (Figure 1) present raw fluxes, not normalized fluxes). If I understand this paragraph correctly, 1) you ranked daily fluxes in reverse order based on magnitude, from largest to smallest 2) you binned these ranked fluxes based on their contribution to total annual emissions. Bin 1 (75% High Cumulative Contribution (HCC)), Bin 2 (50% HCC) and Bin 3 (30% HCC) contain your ranked daily fluxes cumulatively representing 75%, 50%, and 30% of annual emissions, respectively. I initially thought Bin 4 (50% Low Cumulative Contribution (LCC)) was the collection of daily fluxes not included in Bin 2 (50% HCC), but if 50% LCC represents 85% of measurements, that would seem to suggest that instead it might actually be 25% LCC (i.e., the bin selected from daily fluxes ranked in increasing order to sum to 25% of the total cumulative flux), since 75% HCC contains 15% of all daily fluxes (in which case, Bin 4 is actually the collection of daily fluxes not included in Bin 1). It's also non-intuitive that a large number of daily fluxes are being described both as High Cumulative Contribution and Low Cumulative Contribution (the overlap in the 75% HCC bin and a possibly hypothetical 50% LCC bin). And the "50% LCC" term is not included in the Results, Discussion, or figures, though the results presented in Figure 2 appear to represent 50% LCC (though again, it may actually be 25% LCC).

Figure 1 also suggests that there are further factual errors in the binning description in the methods. Figure 1 suggests that there are breakpoints at 75%, 50%, and 25% of the annual flux, creating 4 bins of daily fluxes, though a given daily flux may be present in more than one bin. This suggests that there is no 30% HCC bin, though this is not strictly clear. The results section refers sometimes to a 25% HCC bin, and sometimes to a 30% HCC bin. The caption for Figure 1 first refers to panel D as being based on 25% of the total flux, and later as being based on 30% of the total flux. There is no mention of the 25% HCC bin in the methods.

Your bins often overlap with one-another, which is a bit unusual, and the bins aren't structured in a systematic or symmetrical fashion (30% HCC overlaps with the other two HCC bins, 50% LCC and 50% HCC overlap with 75% HCC). There are also sta-

tistical issues with overlapping bins that need to be kept in mind: because the same observation may be in multiple bins, the bins are not independent, and so comparisons between bins violate any assumptions of independence. In the end, if both overlapping and non-overlapping bins are used, it might be helpful to provide a rationale for that structure as opposed to the alternative of non-overlapping bins. Non-overlapping bins would have the added benefit of being able to explicitly evaluate how variation in flux magnitude affects diurnal patterns, something I would argue the current analytical design is not capable of doing because of the overlap in bins. In addition, the current analysis does not investigate the possibility that diurnal variability is absent when emissions are low and if so, why. This is a question that may not be important for estimating annual fluxes, but is relevant to our scientific understanding of diurnal variation in N2O fluxes.

There's also an important statistical issue with using bins having unequal numbers of observations: the standard error is directly proportionate to the sample size. You use comparisons of the error in Beta to argue that there is no diurnal variation during hot moments, but because of the large differences in sample size, it's not surprising that the estimates of Beta in panel a of figure 1 –which is based on all of the measurements (n= 2,017 days)–or in Figure 2, with 85% of all measurements (n > 1,700 days), have lower uncertainty than the estimates of Beta in Figure 1 panel c (n= 55 days, or 6%) or Figure 1 panel d (n = 27 days, or 3%). Bins of equal size would address this issue. If there is a reason to keep unequal bin sizes, it would seem to be important to at least show that bins containing 3% and 6% of measurements centered around the median exhibit clear diurnal variability, and probably good to evaluate the lowest 3% and 6% of measurements given the possibility that diurnal variability patterns break down at both low and high fluxes. It's essential that this analysis be robust, since at the moment it's the central finding being promoted in the manuscript.

Specific comments:

Introduction: The Introduction provides a review of articles that have evaluated diurnal variability and whether a PMT is reliably observed in a laundry-list format. It is not strictly necessary, but the authors might consider whether it would be possible to provide a greater synthesis of the main results of those studies, and present details in a table. This would have the added benefit of freeing up word count to provide more mechanistic context in the introduction, which is needed but currently lacking: specifically, mechanism behind diurnal variation in N2O emissions from agricultural soils, and what causes that diurnal variability to break down during high emission periods. In addition, "Hot moments" is used multiple times in the abstract, but is missing from the introduction. If "hot moments" is going to be used as a key framing device in the abstract, it also needs to be introduced and contextualized in the introduction. Otherwise, it should be removed from the abstract (and keep in mind that "hot moments" is jargon, though very evocative jargon!).

Line 131: It's not clear how positioning sites within 2.25 kilometers of one another addresses possible effects of manure amendments. It would be helpful to have some clarification, including description of what those possible effects would be.

Line 143: if there's a reference for results from this study, please include.

Line 145-6: "In each campaign, both treatments campaigns dairy slurry was applied" needs to be re-written, perhaps delete "both treatments campaigns"

Line 193: change "valve assembles" to "valve assemblies"

Line 214: Change "Flus" to "Flux"

Line 240: It might be helpful to have a rationale for why these fluxes were removed, rather than included as a net zero flux. If any negative fluxes were excluded, a rationale for that would be needed.

Line 261: Open parenthesis is missing

Line 270: this paragraph should be placed after the normalized cumulative daily flux has already been defined, since it is the independent variable in the analyses.

Line 275: insert "of" before "the annual".

Line 276: how does the "cumulative daily flux" differ from the "daily flux" defined at line 257? If it does not, please use the same term for both; I would suggest sticking with 'daily flux.'

Line 276-277: there's a problem with the sentence construction ("data subsets using to the normalized. . . ").

Line 290 and 291: insert commas in the thousands place for 1,093 and 2,017

Line 293: The abbreviation "HCC" has already been introduced

Line 293 and 295: the use of "measurements" here is a little ambiguous. It might be helpful to clarify whether a "measurement" refers to an individual day, or to an individual (sub-daily) flux. Also not sure reporting a mean of 912 provides useful information to the reader if it's a mean of 5 values ranging from 27 to either 2,017 or over 20,000. Reporting the number of measurements in each bin makes more sense.

Lines 313-316: as noted in my major comment 3 above, the possibility that these results may simply be caused by the huge difference in sample size needs to be resolved.

Line 320: I don't think you've quite shown this yet, because 1) of the issue of having different sample sizes for different flux intensities, and 2) the design of your analysis is not quite an investigation of flux intensity - i.e., different flux intensities are not compared (with the possible exception of 50% HCC and 50% LCC). Also, delete "to".

Line 321 and rest of paragraph: I might think of another way of contextualizing what you call "low" here, since you aren't analyzing fluxes that are low in the context of, for example, your mean flux, but relative to the highest ~10% of fluxes.

Line 330 and following: it would be nice to actually see the results you mention in these lines, as well as some analysis and interpretation of them.

Line 336: Emissions of 723 g N2O-N ha-1 day-1 seem to me to be quite high for a

'low' classification, though you do indeed have some very high individual fluxes. But it seems potentially problematic. For example that Laville et al. 1999 - cited in the manuscript as an example of high fluxes with no diurnal variation–observed maximum hourly fluxes of 700 ng N2O-N m-2 s-1, which extrapolates to a daily flux of roughly 600 g N2O-N ha-1 day-1, falling into the "low flux" category of this manuscript.

Line 364: again, not sure your analysis allows you to say this. You could perhaps just say "diurnal variability of N2O emissions is not present during the largest emission events" under the current analysis.

Figure 1: "N-N2O" is used here, but in the text "N2O-N" is used. Units in panel D (1000, 2000) are mislabeled as 000. Worth a quick double/triple check of concentration measurements and flux calculations to ensure the units are correct since the fluxes are on the high end (many measurements at 1.5 to 3 kg N2O-N ha-1 day-1 during pulses). Delete "and" from "These account for and 100%"

Figure 2: These results could alternatively be included, along with daily flux measurements, as a separate panel in Figure 1.

References: It would be helpful to readers if all references are either indented at the first line, or if a carriage return is placed between references. Scheer et al 2012 Plant Soil would be a good addition, as it includes both high-frequency measurements and evaluation of diurnal variability. DOI 10.1007/s11104-012-1197-4

Line 469: wrong location for Cosentino et al. 2012.

―――――――――――――――――

---

## Author Comment (AC1) · 10 Jan 2020

**Response to Anonymous Referee #1 (Referee comments are shown in *Italics*)**

We thank the referee for their valuable comments that will help improve the manuscript. Before responding to the referee's individual comments we want to clarify that the aim of this paper is not to elucidate the mechanisms responsible for the diurnal variability of soil $N_2O$ emissions, but to provide other researchers with knowledge that will increase the accuracy of their annual emissions estimates and inform the optimization of their sampling protocols. The design of soil trace gas emission monitoring experiments is often based on studies of diurnal cycling of emissions made with very limited data that did not capture seasonal or annual variability, and in which 'hot moment' emissions are absent or overlooked. Our goal in this paper is to characterize the temporal patterns of emissions, rather than explain mechanistic cause of those patterns.

Below we address the referee's concerns individually. Corresponding changes will be made to improve the manuscript.

**General comments (1)**

*Authors state several times that this is the first study of N2O flux temporal variability that includes several hot moments. Which isn't really true, is it? There is Luo et al. 2012 (Decadal variability of soil CO2, NO, N2O, and CH4 fluxes at the Hogwald Forest, Germany), and a lot of other studies from that same study site that show temporal variations in N2O. There are also numerous papers from Australia (e.g. Barton et al. 2007 Nitrous oxide emissions from a cropped soil in a semi-arid climate) – although, to be fair, the Barton paper did not really experience what could be called "hot moments". And there is also the Machado et al. 2019 paper (as cited in the current manuscript) that also measured temporal variability during periods that included "hot moments".*

**Response (1)**

We claim that our publications is the first study of $N_2O$ flux temporal variability that includes multiple difficult-to-measure peak emission events (i.e., "hot moments") and estimates the relative contribution of hot moments to cumulative emissions (lines 12 to 15). It is true that occasional larger emissions periods were observed in the publications cited by the reviewer (Barton et al., 2007; Luo et al., 2012; and Machado et al., 2019). The 'high' emissions reported in these publications, however, are approximately two orders of magnitude smaller than those observed in our experiments. The lack of continuous measurement in these studies makes it impossible to determine the importance of these larger flux periods relative to cumulative emissions.

Luo et al. (2012) observed multiple hot moments and estimated their relative contribution to cumulative emissions. This publication states that pulse events that occurred during soil thaw in 1996 and 2006, accounted for 88% and 87% of the total annual emissions, respectively. Luo, et al. (2012) did not attempt to characterize diurnal variability, however, but rather characterized the variability of $N_2O$ emissions at the seasonal, annual and decadal scale. We have not found a

publication in which the temporal variability of $N_2O$ emissions from the Höglwald Forest, Germany have been studied at a sub-daily frequency or with the purpose of providing a sampling time that best represents the mean daily emissions.

**General comments (2):**

*Also, I don't agree that constraining sampling to particular times of the day provide little benefit. Generally, researchers will be sampling during regular working hours (i.e. between 9 am and 5 pm), during those hours, sampling before 10 am will underestimate fluxes (for 50% of annual flux – i.e. see figure 1c), while sampling after 2 pm will overestimate fluxes. So your preferred sampling time should be between 10:30 to 13:30. No? Even figure 1d makes a strong case for not sampling in the afternoon (between 15:00 and 18:00), because it will overestimate fluxes. That being said, I do agree with you that frequency of sampling during hot moments is more important than what time the sampling took place during those hot moments.*
*So, realistically, I don't think that your conclusions are actually substantiated by your own data. I think that your data still backs up previous research that suggests avoiding the afternoons when sampling for determination of N2O fluxes. Not the most novel conclusion, but I think it is still worthwhile.*

**Response (2)**

We do not dispute the conclusions of previous research about the Preferred Measuring Time (PMT) when a diurnal cycle is observed, rather we conclude that it is *much* more important to sample frequently during peak emissions than to sample at a specific time of day. The recommendations of a PMT in previous research, based on limited data engender false confidence that sampling daily at a particular time is sufficient to yield good estimates of daily and cumulative emissions. We emphasize that such confidence is misplaced.

Previous research is generally based on short experiments that did not include periods of significantly high emission. When a diurnal cycle of emissions **is** observed, this previous research concludes that there is a PMT in the late morning. Our results do not refute this conclusion, but more importantly, they show that these periods when there is a PMT are relatively unimportant in terms of estimating cumulative emissions. A much larger fraction of the cumulative emissions occur during high emissions periods when there is no diurnal cycle and no PMT, and during these periods sampling frequently is essential.

High emissions periods (25% HCC), represent only 6% of the total observations but 25% of the cumulative emissions. On average across the three years, the beta coefficient for each hour during the high emissions periods is computed from data gathered on just 27 days (Lines 291-295). One day's worth of measured emissions during peak emissions is 0.93% (i.e., 25%/27) of the cumulative emissions. On the other hand, low emissions represent 50% of the cumulative emissions, and on average, emissions data from 912 days are used to compute the beta coefficients (Lines 291-295). One day's worth of measurements during low emissions (LCC 50%) is only 0.056% (50% / 912) of cumulative emissions. Thus, getting one accurate measurement

during the high emissions periods is more important than measuring 20 times during low emissions periods (the ratio between 0.93% and 0.056% is 16.5).

We conclude that sampling frequently during peak emissions is more important than sampling less frequently (e.g., daily) at a specific time of day.

**General comments (3):**
*Also, I would like to see a bit more in the discussion about why diurnal patterns are less relevant during periods of high emissions. There is very little on what mechanisms of processes actually drive this. In Figure 1c and 1d, there is much higher variability between 0 and 8h (compared with the rest of the day). Any ideas why this might be? Is there less production? Or is it related to climatic conditions at night?*

**Response (3)**
We would like to be able to provide more insight into the mechanisms of the processes driving the observed patterns of emissions. We invested quite a bit of time looking at weather data, soil and moisture data, and emissions, but were unable to find consistent and reliable explanations for observed emissions in our data. Previous research has linked environmental variables (i.e. soil/air temperature, water filled pore space) to $N_2O$ production but also concluded that there are site- and event-specific factors that drive the biological processes (i.e. depth of $N_2O$ production, soil characteristics, available carbon) and that we did not measure.

We did not explore why there is much higher variability during the late night and early morning (hours 0 to 8) than during the rest of the day at 50 and 25% HCC fluxes. According to the right plots in figure 1, panels C and D, fluxes measured from 0 to 8 are usually smaller than the mean daily flux, but the high variability of the fluxes indicates that this is not always the case. We believe climatic conditions to be influential; nonetheless, other variables that we did not measure are needed to draw meaningful conclusions in this regard (see Shurpali, et al. 2016 and Thies et al., 2019).

What we are able to do with our extensive data set is "to evaluate the use of PMTs as a strategy to improve the accuracy of soil $N_2O$ flux estimates or reduce the necessary frequency of flux measurements in highly fertilized crop systems", lines 120 and 121. We too would like to be able to say more about the mechanisms underlying the observed emissions, and although we cannot, we feel that what we have been able to show about the relative importance of sampling daily at a PMT compared to sampling frequently during high emissions periods is important.

**General comments (4):**
*Finally, try to avoid paragraphs that consist of 1 sentence (e.g. lines 159-161).*

**Response (4)**
Thank you. We agree. We will make appropriate changes and try to do better in the future.

**Responses to specific comments**

| Specific comment | Response |
|---|---|
| Lines 36-38. The topic of sampling frequency and quantifying what it means in terms of uncertainty for your cumulative estimate has been covered well in the paper by Barton et al. 2015 (see your own citation list), and should probably be cited here. | Accepted |
| Line 46: I think it would be worthwhile to cite your Table 1 here, because the table does a good job of summarizing some of the literature on timing and existence of diurnal patterns in N2O fluxes. Although, it doesn't seem like there is that much disagreement. Pretty much all of the studies summarized in Table 1 (11 of the 12 that suggest a PMT) suggest avoiding sampling during the afternoon.
That is pretty good consensus in my opinion. | The table will be cited.
Avoid afternoon sampling is accepted but most likely not to be mentioned because it could create confusion (i.e. avoid afternoon sampling = to accurate cumulative emissions estimates) |
| Line 214: "Flux" not "Flus". | Accepted |
| Line 239: wouldn't it make more sense to report your MDF as a flux per hour? Rather than per day? You are measuring flux rates based on 20 min deployment times and are calculating your daily fluxes by integrating the individual flux measurements for that particular day. So it is possible to have some fluxes below the MDF and others above the MDF on the same day.

Line 239: wouldn't it make more sense to report your MDF as a flux per hour? Rather than per day? You are measuring flux rates based on 20 min deployment times and are calculating your daily fluxes by integrating the individual flux measurements for that particular day. So it is possible to have some fluxes below the MDF and others above the MDF on the same day. | We reported all fluxes in the same units to facilitate comparisons within the publication.

Fluxes below the MDF were removed (line 240), deleted.

It is possible to have fluxes below and above the MDF in the same day. Nonetheless the MDF is very small and the chance of fluxes below this size is very small. We believe that most of the fluxes below the MDF are the result of a chamber not closing during the measuring time.

We have observed that a large number of fluxes below the MDF occurred during rain periods when the chambers remain open and others occurred when the chamber did not close due to a mechanical fault.
Differentiating between mechanical fault and fluxes actually below the MDF is not always possible. |

| | |
|---|---|
| Line 266: the "+0" is unnecessary. | Accepted |
| Line 275: "percentage of the annual". | Accepted |
| Line 289: what happened during 2016-17? Having to throw out 48% of the fluxes is not good (or were these removed because they were below MDF?). I think it might be better to differentiate when fluxes were thrown out because of bad data and when they were thrown out because they were below MDF. | Differentiating between a mechanical fault and a flux below the MDF is not always possible.

2016-17 was our first sampling season through the Wisconsin winter. Multiple mechanical failures reduced the availability of high temporal resolution data during this period. The large number of mechanical failures resulted in the design and construction of new automatic soil chambers. Most of the mechanical problems occurred after an unusual event when the experimental field flooded in late January followed immediately by freezing temperatures that coated the equipment and soil surface with a thick layer of ice. |
| Line 318: Are you sure that this is the first? Doesn't the Machado et al. 2019 paper measure diurnal variability in a fertilized agronomic system with hot moments? | Responded previously. |

---

## Author Comment (AC2) · 10 Jan 2020

**Response to Anonymous Referee #2 (Referee comments are shown in *Italics*)**

We thank the referee for catching errors and for his/her valuable comments. The constructive comments will help us to clarify the objectives of this paper and make a stronger conclusion about the usefulness of PMTs as a way to improve the accuracy of cumulative emissions estimates and/or reduce sampling frequency.

Below we address the referee's concerns. Corresponding changes will be made to improve the manuscript.

**General comments (1)**

*I leave the question of editorial suitability to the editor, but I could use a little more convincing*

*that this manuscript wouldn't be better suited to a journal more focused on agriculture, given the manuscript's narrow focus on questions of sampling representativeness of fertilized fields.*

Suitability of our manuscript for this journal is, of course, a question for the editor, but we hope that this judgment was made before the manuscript was sent for review.

*With respect to novelty, previous studies have similarly found that N2O fluxes often do not*

*exhibit diurnal variability, particularly when fluxes are low or high (e.g., Laville et al 1999, Van der Weerden et al. 2013), although to say that these particular studies are much shorter in duration would be an understatement, and the long-term dataset included here is certainly of value.*

We agree that part of what is unique about our analysis is the long-term dataset. We believe that ours is the first study to examine the existence, nature and timing of diurnal cycling of emissions using a multi-year dataset that includes high-frequency monitoring through all four seasons.

Laville, et al. 1999, Van der Weerden, et al. 2013, support that during the high emissions periods there are no PMTs and sampling frequency should increase during these periods to capture emission variability and increase the accuracy of the mean daily and cumulative flux estimates. As noted, previous studies use very short data sets and they also do not estimate the proportion of fluxes represented by the high emissions periods. The reader of previous articles is left without any insight into how important it is to measure accurately during the high emissions periods versus other periods. We remedy this by presenting the relative contribution to the total cumulative emissions of emissions at 4 different levels (i.e. 50% LCC, 75%HCC, 50%HCC, and 25%HCC).

*Incidentally, one interpretation of the results would seem to be that PMTs could be the optimal design in instances where sub-daily sampling is not possible: if PMTs accurately reflect daily emissions during the majority of sampling days, it makes sense to use them. If on high emission*

*days there is no diurnal variation, measuring during a PMT is as good a time as any. Perhaps the real message might be that effort should be made to sample hot moments in higher than daily temporal resolution.*

We agree that sampling at the PMT is as good a time as any, and in instances where sub-daily sampling is not possible, it is preferred. As we wrote in our second response to Reviewer #1, we do not dispute the conclusions of previous research about the Preferred Measuring Time (PMT) when a diurnal cycle is observed, rather we conclude that it is much more important to sample frequently during peak emissions than to sample at a specific time of day. The recommendations of a PMT in previous research, based on limited data engender false confidence that sampling daily at a particular time is sufficient to yield good estimates of daily and cumulative emissions. We emphasize that such confidence is misplaced.

Our results show that measuring the flux accurately during the high emissions periods contributes as much information about the cumulative flux as measuring 20 times during low emissions periods. Another way to frame this is that, in our experimental system, measuring at the PMT rather than at a randomly chosen time of the day, improves the estimate of cumulative flux less than capturing just one or two measurements of the peak events that are easily missed when sampling less than daily. This is the point we hope to make with our article. The figure below (Fig. 1 *Usefulness of PMTs*) provides information about the accuracy and precision that one can expect in estimates of cumulative emissions made using fluxes measured at the PMTs. In this figure, accuracy and precision are explained in a box-plot format, upper and lower box bounds correspond to the first and third quartiles (the 25th and 75th percentiles), whiskers extend to capture all data, and the horizontal bar represents the mean.

The y-axis represents the ratio between the true cumulative emissions (estimated using the high temporal resolution data) and the cumulative emissions estimated using flux measurements made at a PMT. The ratios were computed at four PMTs, 01:00, 11:00, 22:00, and 23:00 (four vertical panels), these PMTs are common for all levels of CC (complete data set, 75% HCC, 50%HCC, and 25%HCC, see figure 1). Within each panel the ratios were computed at 4 levels: using the full data set, 50%LCC, 50%HCC, and 25%HCC. Within each level of CC, the ratios are computed for the full data set, grey boxes, and for the three seasons; yellow, blue, and green boxes for seasons, 2015, 2016-2016, and 2017-2018 respectively. The horizontal dashed line y = 1 represents the maximum achievable accuracy using our sampling system, this is cumulative emissions computed using high temporal measurements (8 or more flux measurements per day).

The second panel of Fig. 1 (*Usefulness of PMTs*) represents the typical practice of sampling during the morning at the 11:00 PMT. During 2016-2017 (blue) this sampling practice would have significantly underestimated the cumulative flux, while in 2017-18 (green) it would have led to a very large overestimate of the cumulative flux. Even using perfect hindsight to sample only the days with the largest fluxes (25% HCC) at this time leads to large errors in 2 of 3 years and around 5% error in the other (i.e., 2015). If sub-daily sampling is not possible, sampling at the PMT is as good a time as any to sample, but the result is potentially large error in the estimated cumulative flux, and the sign and magnitude of the error varies from year to year.

[Figure]

*Figure 1.* **Usefulness of PMTs**

Below is a summary of the information represented in the figure:

- Sampling at the PMTs results in estimates of cumulative emissions that are between -7 to -22%, +5 to -17%, +6 to -6%, and +2 to -10% accuracy for all years when fluxes are measured daily at 01:00, 11:00, 22:00, and 23:00 respectively (grey boxes, full data set).
- The accuracy (difference from one) and precision (length of the box) of the estimated cumulative emissions based on sampling at the PMT decreases according to the contribution to cumulative emissions in the following order: 50% LCC > full data set > 25%HCC > 50% HCC (except for hours 22:00 and 23:00 during which the 50%HCC yields better estimates than 25%HCC)).
- Using the full data set, sampling at 22:00 or 23:00 yields cumulative emission estimates within +2 and -10% accuracy, respectively, when the data from all the three years are averaged (grey boxes), however, large errors result in estimates of annual cumulative emissions.

*A question I'm left with is: how does variation in the sampling timing and frequency during hot moments affect annual flux estimates (e.g., comparing a single daily measurement at the PMT to varying sets of additional, sub-daily measurements)? In other words, some quantification of the downside of using a PMT would be very interesting, and germane to the central question of this study.*

For each of the PMTs (vertical panels) in the figure, the difference between the first box plot (full data set) and the second box plot (50%LCC) is the improvement in precision and accuracy achieved if high emissions periods were sampled at high temporal resolution (8 measurements or more per day) and that the 50%LCC emissions is sampled daily at a PMT. This is based on the fact that the difference between the ¨full data set¨ emissions and the ¨50%LCC¨ emissions are the high emissions and the assumption that sampling at high temporal resolution during any period of time will result in cumulative emissions estimates with zero errors for that period.

**General comments (2)**

*The analyses presented here are focused on the question of sampling representativeness based on an empirical evaluation of when diurnal variation does and does not occur. It does not attempt to explore why and under what conditions diurnal variation breaks down: there's almost no interrogation of process or mechanism. Understanding the degree to which PMTs are representative of daily fluxes is a worthwhile goal, but the presentation here feels thin without substantive investigation of why PMTs are or are not representative. There are many data here that were collected but don't appear to figure at all in the analysis and are not presented as results, in particular depth-resolved soil temperature and moisture observations (Table 1 informs us that some kind of temperature/N2O relationship existed, but no details on that relationship are presented). Seasonal variation in the duration and timing in PMTs is referred to in passing but not actually presented. There is no consideration of soil C or N.*

We would like to be able to provide more insight into the mechanisms of the processes driving diurnal emissions patters and why and under what conditions diurnal variations breaks down. As we responded to Reviewer #1, we spent quite a bit of time investigating this. As you note, we did collect other measurements and we analyzed weather data, depth-resolved soil and moisture data, over different time periods (e.g., seasonal), but were unable to find consistent and reliable explanations for observed emissions. Previous research has linked environmental variables (i.e. soil/air temperature, water filled pore space) to $N_2O$ production, but has also concluded that there are site- and event-specific factors that drive the biological processes (i.e. depth of N2O production, soil characteristics, available carbon) and that we did not measure, but that might provide insight. As Parkin (1987) observed, soil conditions, particularly the concentrations and forms of nutrients, can vary dramatically on a very small spatial scale, making it very difficult to prediction denitrifying conditions from macro-measurements.

*There are a lot of questions I was curious about while reading the manuscript. For example, does diurnal variation in high fluxes depend at all on whether the pulses are related to fertilizer applications, freeze-thaw events, precipitation events, or some other cause? Do organic and inorganic fertilizer applications affect diurnal variation differently (through effects on soil O2 and organic C availability to denitrifiers)? How does (seasonal) variation in the range of diurnal temperature variation affect diurnal variation during high flux events? Even with the three-year data set there may not be enough replication of events to answer some of these questions statistically, and the organic v inorganic question can't be analyzed statistically with the current*

*experimental design, but I would think some quantitatively-informed discussion would be possible, and could be a good way to take advantage of this very nice dataset.*

The reviewer suggests a number of really great questions that we would love to answer. We have looked at several of these hypotheses, but simply do not have enough data to draw defensible conclusions. Even with data over three years, the number of high flux events that occur under reasonably similar conditions (season, soil moisture/temperature, fertilization, etc.) is often just one. As we are still running this experiment, we do hope to be able to address some of these questions as we gather even more data. For example, we plan to investigate the mechanisms leading to peak emissions during spring freeze-thaw cycles. We hope that with 4 years of high temporal resolution $N_2O$ fluxes, weather, soil temp/moisture data, and soil carbon and nitrogen concentrations now being analyzed, as well as $NO_3$ loss below the root zone, that we will be able to say more about the mechanisms controlling these processes.

**General comments (3)**

*I have a number of concerns with the binning approach used.*

*First, the description of the normalized cumulative daily fluxes in line 274 and following is not*

*very clear and could be done much more simply: I'm not sure it is necessary to define a new concept here (It's also not necessary to normalize, especially since the normalization is not actually carried through in the analysis: the results (Figure 1) present raw fluxes, not normalized fluxes). If I understand this paragraph correctly, 1) you ranked daily fluxes in reverse order based on magnitude, from largest to smallest 2) you binned these ranked fluxes based on their*

*contribution to total annual emissions. Bin 1 (75% High Cumulative Contribution (HCC)), Bin 2 (50% HCC) and Bin 3 (30% HCC) contain your ranked daily fluxes cumulatively representing 75%, 50%, and 30% of annual emissions, respectively. I initially thought Bin 4 (50% Low Cumulative Contribution (LCC)) was the collection of daily fluxes not included in Bin 2 (50% HCC), but if 50% LCC represents 85% of measurements, that would seem to suggest that instead it might*

*actually be 25% LCC (i.e., the bin selected from daily fluxes ranked in increasing order to sum to 25% of the total cumulative flux), since 75% HCC contains 15% of all daily fluxes (in which case, Bin 4 is actually the collection of daily fluxes not included in Bin 1). It's also non-intuitive that a large number of daily fluxes are being described both as High Cumulative Contribution and Low Cumulative Contribution (the overlap in the 75% HCC bin and a possibly hypothetical 50% LCC*

*bin). And the "50% LCC" term is not included in the Results, Discussion, or figures, though the results presented in Figure 2 appear to represent 50% LCC (though again, it may actually be 25% LCC). Figure 1 also suggests that there are further factual errors in the binning description in the methods. Figure 1 suggests that there are breakpoints at 75%, 50%, and 25% of the annual flux, creating 4 bins of daily fluxes, though a given daily flux may be present in more than one bin.*

*This suggests that there is no 30% HCC bin, though this is not strictly clear. The results section refers sometimes to a 25% HCC bin, and sometimes to a 30% HCC bin. The caption for Figure 1 first refers to panel D as being based on 25% of the total flux, and later as being based on 30% of the total flux. There is no mention of the 25% HCC bin in the methods.*

*Your bins often overlap with one-another, which is a bit unusual, and the bins aren't structured*

*in a systematic or symmetrical fashion (30% HCC overlaps with the other two HCC bins, 50% LCC and 50% HCC overlap with 75% HCC). There are also statistical issues with overlapping bins that need to be kept in mind: because the same observation may be in multiple bins, the bins are not independent, and so comparisons between bins violate any assumptions of independence. In the end, if both overlapping and non-overlapping bins are used, it might be helpful to provide a*

*rationale for that structure as opposed to the alternative of non-overlapping bins. Non-overlapping bins would have the added benefit of being able to explicitly evaluate how variation in flux magnitude affects diurnal patterns, something I would argue the current analytical design is not capable of doing because of the overlap in bins. In addition, the current analysis does not investigate the possibility that diurnal variability is absent when emissions are low and*

*if so, why. This is a question that may not be important for estimating annual fluxes, but is relevant to our scientific understanding of diurnal variation in N2O fluxes.*
*There's also an important statistical issue with using bins having unequal numbers of observations: the standard error is directly proportionate to the sample size. You use comparisons of the error in Beta to argue that there is no diurnal variation during hot moments,*

*but because of the large differences in sample size, it's not surprising that the estimates of Beta in panel a of figure 1 –which is based on all of the measurements (n= 2,017 days) – or in Figure 2, with 85% of all measurements (n > 1,700 days), have lower uncertainty than the estimates of Beta in Figure 1 panel c (n= 55 days, or 6%) or Figure 1 panel d (n = 27 days, or 3%). Bins of equal size would address this issue. If there is a reason to keep unequal bin sizes, it would seem*

*to be important to at least show that bins containing 3% and 6% of measurements centered around the median exhibit clear diurnal variability, and probably good to evaluate the lowest 3% and 6% of measurements given the possibility that diurnal variability patterns break down at both low and high fluxes. It's essential that this analysis be robust, since at the moment it's the central finding being promoted in the manuscript.*

There is no 30% HCC bin, it is 25% HCC.  This was written incorrectly in several places. The 50% LCC data represent 96% of the observed fluxes.
Sorry for the confusion, I (Jordi Francis) mixed results from analysis at different levels during revision of the manuscript.
Fluxes in the bins of 50% LCC and 50% HCC do not overlap.

We will refer to 50% LCC in line 321 after mentioning '*low emissions*'.
The comment in which we compare overlapping subsets will be removed (lines 314-316).
We performed a seasonal analysis not showed in the publication (line 333), bins had a similar number of fluxes. Because the proportion of peak event fluxes to low fluxes was very low in each of the bins, the importance of the peak events was not captured.

We did not analyze diurnal patterns at the lower emissions level of 50% LCC (94% of the observed data) because these fluxes represent a small proportion of the total cumulative emissions; increasing sampling frequency during these periods is less efficient than increasing sampling frequency during high emissions periods (50% HCC, 6% of the observed time).

**Responses to specific comments**

| Specific comment | Response |
|---|---|
| Introduction: The Introduction provides a review of articles that have evaluated diurnal variability and whether a PMT is reliably observed in a laundry-list format. It is not strictly necessary, but the authors might consider whether it would be possible to provide a greater synthesis of the main results of those studies, and present details in a table. This would have the added benefit of freeing up word count to provide more mechanistic context in the introduction, which is needed but currently lacking: specifically, mechanism behind diurnal variation in N2O emissions from agricultural soils, and what causes that diurnal variability to break down during high emission periods. In addition, "Hot moments" is used multiple times in the abstract, but is missing from the introduction. If "hot moments" is going to be used as a key framing device in the abstract, it also needs to be introduced and contextualized in the introduction. Otherwise, it should be removed from the abstract (and keep in mind that "hot moments" is jargon, though very evocative jargon!). | The concept of 'hot moments' is defined in lines (88 to 96), we will make corrections to include the term 'hot moments' in these lines.

The mechanisms behind $N_2O$ diurnal variability are not the object of study in this publication |
| Line 131: It's not clear how positioning sites within 2.25 kilometers of one another addresses possible effects of manure amendments. It would be helpful to have some clarification, including description of what those possible effects would be. | We will rewrite this lines, we meant that experimental sites have a history of no manure during at least the 3 previous years, but the experimental sites where within close proximity. |
| Line 143: if there's a reference for results from this study, please include. | Future publication, explained before. |
| Line 145-6: "In each campaign, both treatments campaigns dairy slurry was applied" needs to be re-written, perhaps delete "both treatments campaigns" | Accepted |
| Line 193: change "valve assembles" to "valve assemblies" | Accepted |
| Line 214: Change "Flus" to "Flux" | Accepted |

| | |
|---|---|
| Line 240: It might be helpful to have a rationale for why these fluxes were removed, rather than included as a net zero flux. If any negative fluxes were excluded, a rationale for that would be needed. | Chances of fluxes below this value are very small. We believe that most of the fluxes below the MDF are the result of a chamber not closing during the measuring time.

We have observed that a large number of fluxes below the MDF occurred during rain periods when the chamber remain open and many other were the chamber did not close due to mechanical failure. Differentiating between mechanical failure and below the MDF is not always possible. |
| Line 261: Open parenthesis is missing | Close parenthesis will be removed |
| Line 270: this paragraph should be placed after the normalized cumulative daily flux has already been defined, since it is the independent variable in the analyses. | Normalized size flux is not the independent variable in the analysis. Normalized size flux is used to bin fluxes. |
| Line 275: insert "of" before "the annual". | Accepted |
| Line 276: how does the "cumulative daily flux" differ from the "daily flux" defined at line 257? If it does not, please use the same term for both; I would suggest sticking with 'daily flux.'
Line 276-277: there's a problem with the sentence construction ("data subsets using to the normalized. . . "). | Lines 275 to 279 will be re-written |
| Line 290 and 291: insert commas in the thousands place for 1,093 and 2,017 | Accepted |
| Line 293: The abbreviation "HCC" has already been introduced | Accepted |
| Line 293 and 295: the use of "measurements" here is a little ambiguous. It might be helpful to clarify whether a "measurement" refers to an individual day, or to an individual (sub-daily) flux. Also not sure reporting a mean of 912 provides useful information to the reader if it's a mean of 5 values ranging from 27 to either 2,017 or over 20,000. Reporting the number of measurements in each bin makes more sense. | Accepted |

| | |
|---|---|
| Lines 313-316: as noted in my major comment 3 above, the possibility that these results may simply be caused by the huge difference in sample size needs to be resolved. | Answered previously, will be removed |
| Line 320: I don't think you've quite shown this yet, because 1) of the issue of having different sample sizes for different flux intensities, and 2) the design of your analysis is not quite an investigation of flux intensity - i.e., different flux intensities are not compared (with the possible exception of 50% HCC and 50% LCC). Also, delete "to". | Answered previously |
| Line 321 and rest of paragraph: I might think of another way of contextualizing what you call "low" here, since you aren't analyzing fluxes that are low in the context of, for example, your mean flux, but relative to the highest _10% of fluxes. | Accepted, low will be replaced by 50% LCC |
| Line 330 and following: it would be nice to actually see the results you mention in these lines, as well as some analysis and interpretation of them. | Answered previously |
| Line 336: Emissions of 723 g N2O-N ha-1 day-1 seem to me to be quite high for a 'low' classification, though you do indeed have some very high individual fluxes. But it seems potentially problematic. For example that Laville et al. 1999 - cited in the manuscript as an example of high fluxes with no diurnal variation–observed maximum hourly fluxes of 700 ng N2O-N m-2 s-1, which extrapolates to a daily flux of roughly 600 g N2O-N ha-1 day-1, falling into the "low flux" category of this manuscript. | Noted, peak fluxes in this manuscript are much larger than those recorded by Laville et al. 1999. |
| Line 364: again, not sure your analysis allows you to say this. You could perhaps just say "diurnal variability of N2O emissions is not present during the largest emission events" under the current analysis. | Accepted |

| | |
|---|---|
| Figure 1: "N-N2O" is used here, but in the text "N2O-N" is used. Units in panel D (1000, 2000) are mislabeled as 000. Worth a quick double/triple check of concentration measurements and flux calculations to ensure the units are correct since the fluxes are on the high end (many measurements at 1.5 to 3 kg N2O-N ha-1 day-1 during pulses). Delete "and" from "These account for and 100%" | Accepted |
| Figure 2: These results could alternatively be included, along with daily flux measurements, as a separate panel in Figure 1. | We believe this will make the figure hard to read (cramped plots and small font size). |
| References: It would be helpful to readers if all references are either indented at the first line, or if a carriage return is placed between references. Scheer et al 2012 Plant Soil would be a good addition, as it includes both high-frequency measurements and evaluation of diurnal variability. DOI 10.1007/s11104-012-1197-4 | Accepted |
| Line 469: wrong location for Cosentino et al. 2012. | Accepted |